# Impact of the Acceptance of the Recommendations Made by a Meropenem Stewardship Program in a University Hospital: A Pilot Study

**DOI:** 10.3390/antibiotics11030330

**Published:** 2022-03-02

**Authors:** Jorge Alba Fernandez, Jose Luis del Pozo, Jose Leiva, Mirian Fernandez-Alonso, Irene Aquerreta, Azucena Aldaz, Andres Blanco, Jose Ramón Yuste

**Affiliations:** 1Infectious Diseases Department, Hospital Universitario San Pedro, 26006 Logroño, Spain; jalbaf@riojasalud.es; 2Infectious Diseases Division, Clínica Universidad de Navarra, 31008 Pamplona, Spain; jdelpozo@unav.es (J.L.d.P.); ablancod@unav.es (A.B.); 3Microbiology Division, Clínica Universidad de Navarra, 31008 Pamplona, Spain; jleiva@unav.es (J.L.); mferalon@unav.es (M.F.-A.); 4Instituto de Investigación Sanitaria de Navarra (IdiSNA), 31008 Pamplona, Spain; 5Pharmacy Division, Clínica Universidad de Navarra, 31008 Pamplona, Spain; iaquerreta@unav.es (I.A.); aaldaz@unav.es (A.A.); 6Department of Internal Medicine, Clinica Universidad de Navarra, 31008 Pamplona, Spain

**Keywords:** carbapenem, ertapenem, antimicrobial stewardships

## Abstract

Antimicrobial stewardship programs (ASP) promote appropriate antimicrobial use. We present a 4-year retrospective study that evaluated the clinical impact of the acceptance of the recommendations made by a meropenem-focused ASP. A total of 318 meropenem audits were performed. The ASP team (comprising infectious disease physicians, pharmacists and microbiologists) considered meropenem use in 96 audits (30.2%) to be inappropriate. The reasons to consider these uses inappropriate were the possibility of de-escalating to a narrower-spectrum antibiotic, in 66 (68.7%) audits, and unnecessary meropenem use, in 30 (31.3%) audits. The ASP team recommended de-escalation in 66 audits (68.7%) and discontinuation of meropenem in 30 audits (31.3%). ASP interventions were stratified according to whether or not recommendations were followed. The group in which recommendations were accepted and followed (i.e., accepted audit, AA) included 66 audits (68.7%) and the group in which recommendations were not followed (i.e., rejected audit, RA) included 30 (31.3%) audits. The comorbidity of the AA group (Charlson score) was higher than in the RA group (7.0 (5.0–9.0) vs. 6.0 (4.0–7.0), *p* = 0.02). Discontinuation of meropenem was recommended in 83.3% of audits in the AA group vs. 62.2% in the RA group (OR 3.05 (1.03–8.99), *p* = 0.04). Ertapenem de-escalation resulted in a 100% greater rate of follow-up compared with the non-carbapenem option (100% vs. 51.9%, OR 1.50 (1.21–1.860), *p* = 0.001). Significant differences were observed in the AA group when cultures were taken before antibiotic prescription—98.5% vs. 83.3% (*p* = 0.01, OR 13.0 (1.45–116.86))—or when screening cultures were taken—45.5% vs. 19.2% (*p* = 0.03, OR 3.5 (1.06–11.52)). There were no differences between the groups in terms of overall mortality and 30-day mortality, length of stay, *Clostridiodes difficile* infection, 30-day readmission or hospitalization costs. In conclusion, meropenem ASP recommendations contributed to a decrease in meropenem prescription without worsening clinical and economic outcomes.

## 1. Introduction

Antibiotics are one of the most common drugs prescribed in healthcare facilities [1]. It is estimated that more than 40% of prescriptions do not follow local guidelines, or are not adjusted to microbiological culture results [2,3]. Antibiotics are singular drugs, whose effect on bacterial ecology and misuse can directly impact microbial resistance [4], thereby increasing morbidity and hospital costs. It has been estimated that broad-spectrum antimicrobial misuse results in more than 23,000 deaths and $20 billion in costs every year in the United States [5,6].

Antimicrobial stewardship programs (ASPs) are ways to reduce the inappropriate use of antibiotics. There are many forms that ASPs can take. The form most recommended by societies such as the Infectious Diseases Society of America is a prospective audit and feedback (PAF) process, in which a prescription is revised by a multidisciplinary team, which includes infectious disease (ID) specialists, clinical microbiologists and hospital pharmacists. The main goals of ASPs are to improve healthcare quality by optimizing antibiotic prescriptions, to reduce antimicrobial resistance and to conduct medical training on antibiotic use [7,8,9].

Meropenem is a broad-spectrum class II carbapenem used in nosocomial infections as an empiric treatment because it is active against the most common hospital-acquired resistant microorganisms [10]. During the last several years, its effectiveness has been compromised because of the emergence of resistances [11].

In this study, we present the patient characteristics and results of a 4-year retrospective meropenem ASP in a third level university hospital in Spain.

## 2. Results

Our ASP team conducted 2367 audits between March 2014 and February 2018. Four audits were excluded in the study due to the patients either being pregnant or under 18 years old. 318 of these audits (13.4%) were related to a meropenem prescription. The ASP team considered meropenem to be an inappropriate prescription in 96 (30.2%) audits. In 66 (68.7%) audits, the recommendation was accepted (AA) and in 30 (31.3%) it was rejected (RA). (Figure 1).

### 2.1. Demographics and the Basal Clinical Characteristics of Patients

The main characteristics of the patients audited are shown in Table 1. Both groups had a high Charlson median score (6.0 (5.0–8.0)), but this was higher in the AA group (7.0 (5.0–9.0) vs. 6.0 (4.0–7.0)), with a statistical difference found, *p* = 0.02.

All audited patients with inappropriate meropenem use had at least one risk factor to develop a multiresistant bacterial infection, which was being in empiric therapy with meropenem. The most frequent related factors were the use of antibiotics in the last 30 days and inpatient treatment in the last 60 days (59.4% and 56.5%, respectively); we found no statistical differences between the groups.

### 2.2. Microbiological Data

Site infection cultures were taken before meropenem was prescribed in 90 (93.4%) audits (98.5% in the AA group vs. 83.3% in the RA group, OR 13.0 (1.45–116.86), *p* = 0.01). The cultures were positive in 49 (54.4%) audits, with 36 (54.5%) in the AA group and 13 (52.0%) in the RA group, and no statistically significant difference between the groups. Gram-negative bacteria (GNB) were isolated in 36 (73.5%) of the positive cultures but ESBL/AmpC production resistance was only identified in four (11.1%) of them. There was no isolation of carbapenem-resistant pathogens.

Screening cultures (nasal, inguinal and perianal) were taken in 59 (61.5%) audits, with the isolation of enterobacteria with ESBL, ampC or carbapenem-resistance mechanism in 20 (33.9%) of them. ESBL enterobacteria were located in 12 (60.0%) patient screening cultures, ampC b-lactamase-producing enterobacteria in six (30.0%) and a carbapenem-resistant mechanism in two (10.0%). A significant difference was only found when comparing positive screening cultures (15/33 (45.5%) in AA vs. 5/26 (19.2%) RA, OR 3.5 (1.06–11.52), *p* = 0.03) (Table 2).

### 2.3. Intervention Data

The most frequent reason to consider a prescription inappropriate was the excessively broad use of meropenem in 66 audits (68.7%), followed by unnecessary meropenem use, which was judged to have occurred in 30 audits (31.3%). In unnecessary meropenem use, a statistical difference was found in favor of the AA group (83.3% vs. 62.2%, OR 3.05 (1.03–8.99), *p* = 0.04). There was no incorrect antibiotic prescription or dosage change recommendation in any case. The most frequent intervention recommended by the ASP team was de-escalation, which was made in 66 audits (68.7%), followed by the discontinuation of meropenem, which occurred in 30 audits (31.3%). Where the discontinuation of meropenem was recommended, a statistical difference in favor of the AA group was found (83.3% vs. 62.2%, OR 3.05 (1.03–8.99), *p* = 0.04). See Table 3 and Figure 2 for details.

The recommended option for de-escalation was ertapenem (class I carbapenem) in 14 (21.2%) audits. The recommendation was accepted in 100% of these audits. There was a significant association in AA compared to a non-carbapenem option—14/14 (100%), OR 1.50 (1.21–1.86), *p* = 0.001. See Table 3 for details.

### 2.4. Outcomes

A clinical cure at the end of meropenem treatment was achieved in 84 (87.5%) audits. A total of 12 (12.5%) audited patients died, and four (33.3%) of these deaths were related to an infection. The 30-day mortality ratio was six (6.3%), and two of these deaths (33.3%) were related to an infection in the AA group. Readmission 30 days after hospital discharge was needed in 26 (31%) of the audited patients, and 16 (61.6%) of these readmissions were related to an infection. There were no statistically significant differences between the groups.

The median length of stay was 16.0 (10.3–28.5) days. The length of stay after ASP evaluation was 6.0 (3.0–14.0) days. A superinfection with *Clostridioides difficile* was isolated in seven (7.3%) audits after ASP, but no statistically significant differences between the groups were found.

In terms of economic and consumption outcomes, the median defined daily doses (DDDs) per 1000 patients per day was 0.042 (0.032–0.058), with 0.037 in the AA group and 0.045 in the RA group. The median DOTs per 1000 patients per day was 0.034 (0.026–0.043), with 0.030 in the AA group and 0.043 in the RA group. A difference was found in favor of the AA group (*p* < 0.001) in both cases. Total hospitalization costs reached a median of €17,077.0 (€9320.3–€34,261.3) and exhibited no statistically significant difference between the groups, though lower costs were observed in the AA group (€14,903.0) compared with the RA group (€18,412.9). See Table 4 for details.

## 3. Discussion

ASP is a useful tool to lower carbapenem pressure in selected populations, such as patients with high comorbidity, while maintaining clinical and economic outcomes. In this study we can see that both groups had a high Charlson score (median > 6.0), which was even higher in the AA group compared to the RA group (7.0 vs. 6.0, *p* = 0.02) [12,13,14,15]. This difference may be explained because when the Charlson data were analyzed, the only category in which differences were found was the “malignant tumor with metastasis” category, which accounts for six points by itself. The acceptance or rejection of the ASP recommendation did not affect the clinical outcomes of the patients in terms of infection-related mortality (i.e., 30-day mortality or 30-day infection-related mortality), 30-day readmission or length of stay. In a study, Merino et al. [16] compared pre-ASP vs. post-ASP period patients in terms of bloodstream infections. They found a statistical difference in the post-ASP period in those with a high Charlson index (>3) in areas such as better infection source control and optimal definitive antimicrobial therapy (104 vs. 82, *p* = 0.018). These results suggest that ASP can also be a useful and safe tool in patients with high comorbidity.

Prior to the start of meropenem, more infection site cultures were taken in the AA group than in the RA group (65 audits or 98.5% vs. 25 audits or 83.3%, *p* = 0.01). This could be explained by the fact that physicians in charge are more receptive to a recommendation if there are infection site cultures taken, even if no pathogen is isolated. Infection site cultures are a basic tool in order to choose a more specific antibiotic and de-escalate [17,18], and it is also important to take them before the antibiotic is started to increase its rentability [19]. Although there are studies where antibiotic adjustment has been associated with worse outcomes, such as increased length of stay [20], an antibiotic adjustment (mainly de-escalation) is endorsed by many societies, such as the European Society of Intensive Care Medicine and the European Society of Clinical Microbiology and Infectious Diseases [21].

The rate of follow-up on audit recommendations was greater for patients in whom a screening culture had been performed (15/33 or 45.5% in the AA group vs. 5/26 or 19.2% in the RA group, *p* = 0.03), regardless of whether there was an isolation or not. Furthermore, almost all cultures (19/20, 95%) were inguinal or perianal. These kinds of cultures are more associated with an ESBL and/or AmpC resistant microorganism [22] and, therefore, a carbapenem empiric prescription can be justified. The relationship between antibiotic prescription and colonization with multidrug-resistant bacteria has been widely described [23]. Giannella et al. [24] even developed a score to predict mortality in patients with bloodstream infections due to carbapenem-resistant Enterobacteriaceae-colonized patients. Furthermore, other studies, such as Veherschild et al. [25], have reported that, in hematological patients, a previous colonization with ESBL bacteria is an important risk factor for bloodstream infection with ESBL bacteria, OR 52.00 (5.71–473.89). Therefore, screening cultures may be a useful tool for assisting an ASP in selecting an appropriate antimicrobial regimen (or downgrading one) and the physician in charge may choose to follow a recommendation based on these cultures in certain patients.

In cases where meropenem was judged unnecessary, the physician in charge tended to be more likely to follow up on the audit’s recommendation to discontinue than they were in cases where meropenem was judged to be excessively broad spectrum (83.3%, OR 3.05 (1.03–8.99), *p* = 0.04) (see Figure 2). Rusell et al. [26] retrospectively analyzed meropenem use in 107 inpatients. In 14 of them, a meropenem recommendation was made, and in eight patients (60%), the recommendation was to discontinue its use. Similar results were shown by Conlon-Bingham et al. [27] in a quasi-experimental study in which 75 patients treated with meropenem were reviewed. The ASP recommended discontinuation of meropenem in 47% of patients. In their opinion, acceptance by the physician in charge may be explained because the median treatment duration for meropenem is 8 days, and in most infections 7–10 days of therapy are usually recommended [28]). In our study, a significant difference (*p* = 0.001) was also found when a recommendation included de-escalation to a narrower carbapenem (ertapenem) rather than to other antibiotics (100% AA vs. 51.9% RA). It is possible that a physician in charge is more likely to follow a recommendation that includes a carbapenem option than to switch to another antibiotic. This indicates that in some circumstance, such as when an ASP member knows how the physician in charge tends to respond to the ASP recommendations, it could be better to recommend an incomplete de-escalation (i.e., to ertapenem) knowing that the physician will follow it, rather than recommend a more complete and correct de-escalation (i.e., to amoxicillin), which the physician in charge is less likely to follow.

The consumption of meropenem presented an average decrease of 0.017 DDDs between the AA and RA groups (0.037 vs. 0.054, *p* < 0.001). DOTs per 1000 patients per day were 0.034 (0.026–0.043), with 0.030 in the AA group and 0.043 in the RA group (*p* < 0.001), without any change in clinical outcomes (mortality, *Clostridioides difficile* infection, length of stay or readmission). This has been observed in other similar studies. U. Ni Riain et al. [29] conducted a study examining an ASP’s recommendations about meropenem when these were followed and not followed. They found a decrease of more than 9 days (4.5 vs. 14.0 days) of treatment when the ASP made a recommendation which was followed, which in turn led to a decrease in meropenem consumption. Yea-Yuan Chang et al. [30] found a reduction of 34.9% in carbapenem consumption when comparing a pre-ASP vs. post-ASP period. Furthermore, they also found a reduction in *Acinetobacter baumannii* carbapenem resistance.

Finally, although we did not find statistically significant cost savings, we obtained overall cost savings of more than €3500 (€18,412–€14,903) per patient (*p* = 0.51). If we take these results and multiply them by all of the patients (66) we could achieve more than €230,000 in savings over 4 years. Our results are similar to those of JF García-Rodriguez et al. [31], who estimated cost savings of €837,936 in a followed vs. not followed ASP study, though these authors did not find significant differences either.

Whilst this study has some limitations, such as its use of only a single center, its observational design, case-control and the fact its data were analyzed retrospectively, it shows that a meropenem ASP can be a useful tool without compromising clinical or economic outcomes even in high comorbidity patients.

## 4. Materials and Methods

This was a retrospective study conducted from 1 March 2014 to 28 February 2018 in the Clinica Universidad de Navarra (Pamplona, Spain) a 300-bed university medical center.

Our ASP team is composed of a multidisciplinary group that includes infectious disease (ID) specialists, clinical microbiologists and hospital pharmacists. Meropenem prescription audits are among the many kinds of audits made daily. In our hospital, pharmacists identify potential instances of inappropriate meropenem prescriptions and present these to other members of the ASP to discuss whether it is an appropriate prescript or not, according to local guidelines. If it is not, a recommendation is made that the physician in charge is free to follow or not. 2 days later, the audit is revised in order to know if the recommendation was followed or not. If one patient underwent more than one audit per admission, only the first audit was evaluated.

This is a case-control study where cases are defined as having accepted the recommendations of the audit (AA) and controls as having rejected the recommendations of the audit (RA). For this study, patients who were pregnant women and patients under 18 years old were excluded. The follow-up was conducted up to 30 days after discharge and data were collected using the informatic ASP tool of the Clinica Universidad de Navarra informatic clinic tool.

The demographics and clinical variables of the patients evaluated were age, gender, and the severity of the underlying illness defined by the Charlson comorbidity score [32]. Other variables comprised the following: the presence of bacteremia; the department in charge (medical or surgical); the place of the acquisition of the infection (community, related to healthcare or nosocomial); infectious syndrome (respiratory, abdominal, urinary, skin and soft tissue, bone and joint and primary); and risk factors for multiresistant infection (i.e., being an inpatient for more than 48 h in previous 2 months, being an inpatient in a healthcare facility during the previous 2 months or having received an antibiotic prescription in the previous 30 days).

Microbiological variables collected included cultures taken from infection sites and isolation of Gram-negative bacteria, with any special resistances (ESBL, AmpC and/or carbapenem resistance) recorded where these were observed. In cases where two or more cultures were positive with the same organism, only the infectious site culture that was most representative of the infectious syndrome was recorded. Screening cultures, when available, were recorded if there was an ESBL, AmpC or carbapenem-resistant bacteria, and its localization was also included; the localizations could be nasal (cutaneous colonization), inguinal and perianal (digestive colonization). *Clostridioides difficile* infections were also investigated after the audit.

The clinical outcomes included the following: the total length of stay (defined as median days); the length of stay after the ASP audit; the median number of days of meropenem treatment before the ASP evaluation; clinical cure at the end of meropenem treatment (clinically defined as an improvement in the general condition and symptoms related to the infection syndrome, the absence/improvement of fever and the improvement of inflammatory tests, such as procalcitonin, or C-reactive protein, CRP); mortality (infection-related and non-infection-related, defined by the physician in charge depending on evidence of infection, i.e., persistent high inflammatory tests, fever, positive cultures and evidence of death by other conditions, such as neoplasm); 30-day mortality; and readmission rates. The ASP evaluation (i.e., unnecessary or excessively broad spectrum) and its recommendation (discontinuation and de-escalation), days of treatment with meropenem (DOTs) and antibiotic consumption (defined as DDD per 1000 patients per day [33]) were also analyzed.

Economic data were defined as global hospitalization costs adjusted to the Spanish Harmonised Index of Consumer Prices (HICP) to the year 2021.

For the statistical analysis, continuous variables were expressed as means and their standard deviations, or as medians and their interquartile ranges. For the analysis of the quantitative variables, an exploratory analysis was carried out, as well as parametric tests (Student’s *t*-test or Welch’s *t*-test according to the homogeneity of the variances) or non-parametric tests (Mann–Whitney’s or the U of the median as a function of whether the distributions are equal or not), depending on the normality or lack thereof in the distribution of the data. Qualitative variables were analyzed in N × M contingency tables using Pearson’s Chi-squared test with standardized adjusted residuals analysis. IBM SPSS 23.0 was used for the analysis and a *p* value < 0.05 was considered significant.

## 5. Conclusions

In patients with high comorbidity, the recommendations of a meropenem antimicrobial stewardship program reduced the consumption of meropenem without compromising clinical factors (mortality, readmission, length of stay and infection with *Clostridioides difficile*) and without increasing hospitalization costs.

## Figures and Tables

**Figure 1 antibiotics-11-00330-f001:**
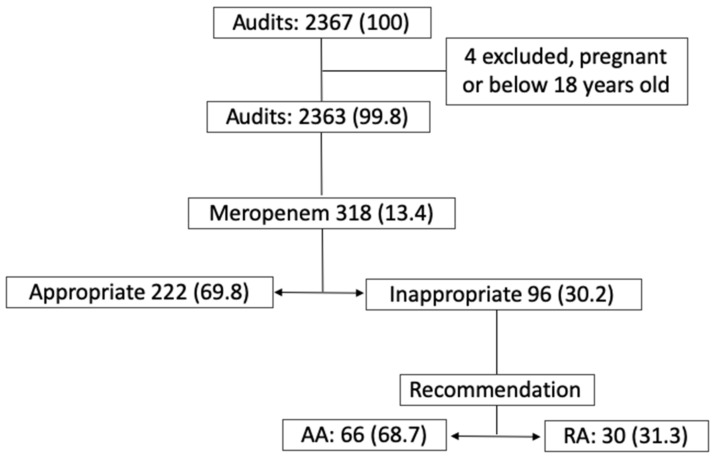
Antimicrobial Stewardship Program briefing, no. (%).

**Figure 2 antibiotics-11-00330-f002:**
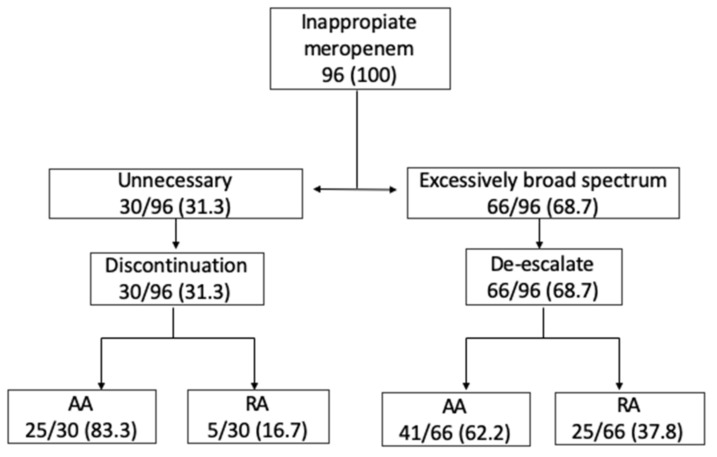
Antimicrobial Stewardship Program meropenem intervention and recommendation, no. (%).

**Table 1 antibiotics-11-00330-t001:** Demographics and clinical characteristics in the admission of patients with inappropriate meropenem prescriptions according to the follow-up on the recommendations conducted by the ASP team.

	Total *N* = 96	Accepted Audit (AA) *N* = 66	Rejected Audit (RA) *N* = 30	*p* Value
Mean age, ± SD, years (range)	67.5 ± 13.0 (39–98)	66.6 ± 13.0 (39–92)	69.5 ± 12.9 (42–98)	0.31
Male Gender, no. (%)	65 (67.7)	42 (63.6)	23 (76.7)	0.21
Charlson index, median (Q1–Q3)	6.0 (5.0–8.0)	7.0 (5.0–9.0)	6.0 (4.0–7.0)	0.02
Department in charge, surgical, no. (%)-	8 (8.3)	8 (12.2)	0 (0)	0.06
Place of acquisition, no. (%)				
Community	16 (16.7)	9 (13.6)	7 (23.3)	0.23
Heatlhcare related	28 (29.1)	18 (27.3)	10 (33.3)	0.63
Nosocomial	52 (54.2)	39 (59.1)	13 (43.3)	0.15
Bacteriemia, no. (%)	1 (1.0)	1 (1.5)	0 (0)	1
Infectious syndrome, no. (%)				
Respiratory	39 (40.6)	24 (36.3)	15 (50.0)	0.21
Abdominal	24 (25.0)	18 (27.3)	6 (20.0)	0.44
Urinary	10 (10.4)	9 (13.6)	1 (3.3)	0.16
Skin and soft tissue	7 (7.3)	6 (9.0)	1 (3.3)	0.72
Bone and joint	3 (3.2)	2 (3.0)	1 (3.3)	0.68
Primary	13 (13.5)	7 (10.6)	6 (20.0)	0.14
Risk factors for MR * infection, no. (%)				
Inpatient > 48 h last 2 months	54 (56.3)	36 (54.4)	18 (60.0)	0.61
Healthcare facility last 2 months	6 (6.3)	4 (6.1)	2 (6.7)	1
Antibiotic lasts 30 days	57 (59.4)	37 (56.1)	20 (66.7)	0.32
Admission days until ASP, median (Q1–Q3)	8.0 (5.0–13.0)	8.5 (5.0–13.0)	6.5 (4.0–11.3)	0.27
Meropenem DOT ^&^ until ASP, median (Q1–Q3)	5.0 (4.0–7.8)	5.0 (4.0–8.0)	5.0 (3.8–6.3)	0.7
Total meropenem DOT, median (Q1–Q3)	8.0 (6.0–10.0)	7.0 (5.0–9.0)	10.0 (7.8–15.3)	<0.001

* MR: Multiresistant; **^&^** DOT: Days of therapy.

**Table 2 antibiotics-11-00330-t002:** Microbiological data.

	Total	Accepted Audit (AA)	Rejected Audit (RA)	*p* Value
Infection site culture prior to meropenem, n°/total audits (%)	90/96 (93.4)	65/66 (98.5)	25/30 (83.3)	0.01
Isolation, n°/total cultures (%)	49/90 (54.4)	36/65 (54.5)	13/25 (52.0)	0.77
Gram-negative bacterial isolation —cases/total isolation—(%)	36/49 (73.5)	25/36 (69.4)	11/13 (84.6)	0.46
Special resistance mechanism, n°/total BGN isolation (%)	4/36 (11.1)	2/25 (8.0)	2/11 (18.2)	1
ESBL	3/4 (75.0)	1/2 (50.0)	2/2 (100.0)	1
AmpC	1/4 (25.0)	1/2 (50.0)	0/2 (0)	1
Carbapenem resistance	0/4 (0)	0/2 (0)	0/2 (0)	-
Screening cultures’ isolation, n°/total taken (%), *at least one*	20/59 (33.9)	15/33 (45.5)	5/26 (19.2)	0.03
Nasal	1/20 (5.0)	1/15 (6.7)	0 (0)	-
Inguinal	12/20 (60.0)	8/15 (53.3)	4/5 (80.0)	0.36
Perianal	20/20 (100)	15/15 (100.0)	5/5(100.0)	1.0
Screening cultures’ resistance, n°/total taken (%), *at least one*	20/59 (33.9)	15/33 (45.5)	5/26 (19.2)	0.03
ESBL	12/20 (60.0)	11/15 (73.3)	1/5 (20.0)	0.12
AmpC	6/20 (30.0)	3/15 (20.0)	3/5 (60.0)	0.07
Carbapenem resistance	2/20 (10.0)	1/15 (6.7)	1/5 (20.0)	0.44

**Table 3 antibiotics-11-00330-t003:** Antimicrobial Stewardship Program intervention data.

	Accepted Audit	Rejected Audit	*p* Value
Reason and recommendation of meropenem audit: Unnecessary/Discontinuation of meropenem (UM), n°/UM (%) Excessively broad spectrum/De-escalate (OS), n°/OS (%) *TOTAL*, *n*°*/total (%)*—	25 (83.3) 41 (62.2) 66 (68.7)	5 (16.7) 25 (37.8) 30 (31.3)	0.04
De-escalation recommedation: Carbapenem, ertapenem (CE), n°/CE (%) Non-Carbapenem (NC), n°/NC (%) *TOTAL*, *n*°*/total (%)*	14 (100) 27 (51.9) 41 (62.2)	0 (0) 25 (48.1) 25 (37.8)	0.001

**Table 4 antibiotics-11-00330-t004:** Clinical, consumption and economic outcomes.

	Total *N* = 96	Accepted Group *N* = 66	Rejected Group *N* = 30	*p* Value
Clinical outcomes				
Inpatient mortality, n°/audits (%)	12/96 (12.5)	11/66 (16.4)	1/30 (3.3)	0.08
Infection-related	4/12 (33.3)	3/11 (27.3)	1/1 (100)	0.33
30-day mortality, n°/audits-mortality (%)	6/84 (7.1)	4/55 (7.3)	2/29 (6.9)	1.0
Infection-related, n°/30-day mortality	2/6 (33.3)	2/4 (50.0)	0/2 (0)	0.53
30-day readmission, n°/audits-mortality (%)	26/84 (31.0)	18/55 (32.7)	8/29 (27.6)	0.80
Infection-related, n°/30-day readmission	16/26 (61.6)	12/18 (66.7)	4/8 (50.0)	0.66
Clinical cure at end of meropenem treatment, n°/audits (%)	84/96 (87.5)	55/66 (83.3)	29/30 (96.7)	0.1
Total length of stay, median (Q1–Q3) (days)	16.0 (10.3–28.5)	15.0 (11.0–29.5)	17.0 (9.0–27.5)	0.89
Length of stay after audit-median (Q1–Q3) (days)	6.0 (3.0–14.0)	6.0 (2.0–11.8)	7.0 (3.0–17.8)	0.34
CD infection, n°. (%)	7 (7.3)	4 (6.1)	3 (10.0)	0.37
Carbapenem-resistant organism within 30 days of the start of carbapenem therapy, n°. (%)	0 (0)	0 (0)	0 (0)	-
Economic and consumption outcomes				
DDD meropenem per 1000 patient-day, median (Q1–Q3) (days)	0.042 (0.032–0.058)	0.037 (0.030–0.051)	0.054 (0.039–0.091)	<0.001
DOT meropenem per 1000 patient-day, median (Q1–Q3) (days)	0.034 (0.026–0.043)	0.030 (0.022–0.039)	0.043 (0.034–0.066)	<0.001
Hospitalization charges, median (Q1–Q3), €	17,077.0 (9320.3–34,261.3)	14,903.0 (8995.1–34,285.0)	18,412.9 (9947.4–35,449.2)	0.51

## Data Availability

All data is available and stored in the informatic tool of our hospital, Clínica Universidad de Navarra.

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
