# Peer review of "Impact of the Acceptance of the Recommendations Made by a Meropenem Stewardship Program in a University Hospital: A Pilot Study"

_antibiotics, 2022, doi:10.3390/antibiotics11030330_

Round 1
Reviewer 1 Report
The study tried to evaluate the clinical and economic impact of a meropenem-focused ASP. ASP is one of the most effective approach to confine antimicrobial resistance, and the paper covered an important element in confining antimicrobial resistance, especially in the hospital setting. However, there are some issues needed to be addressed.
- A clear definition of the ASP is required. The study was designed as a retrospective case control study. The exposure was whether the physician accepted the audit recommendations. However, this design could not answer the question about the impact of the ASP. Because both groups were under this policy (ASP), no matter the physician accept the recommendation or not. Then under this design, the study would answer the question about the impact of the physician’s acceptance of the recommendations on the patient outcomes. This made the study focus turns to physicians’ performance instead of the policy itself. Normally studies on policy evaluation would adopt designs such as pre- and post- design as the paper stated (line 146), or a quasi-experimental design (Ahmed A. Sadeq, et al, https://doi.org/10.3390/antibiotics10111289, a similar study published last year). The current design was not power enough to measure the impact of the policy itself when both groups were part of the policy.
- The sample size calculation was not provided. I doubt the non-significance between the two groups in the demographic and clinical characteristics was due to the low sample size.
- I’d recommend to refer STROBE guideline for each session, especially for the Method session.
- Typos were found (for instance, figure 2; line 278). The writings should be checked.
Reviewer 2 Report
Thank you for your article. Interesting theme. Despite the relative low resistance rates, carbapenem use is high, and stewardship interventions seem to have addressed some of the overuse / misuse of carbapenems.
- The structure of the article could have been re-organized. It would have been easier to read the methods first, prior to results or discussion. It was confusing to read the results, then review the methods.
2. The antibiotic stewardship concept in the introduction can be revised. It is well known that prospective audit and feedback (PAF) is a key stewardship strategy as discussed by Dellit's paper published in CID in 2007. I am not sure if we need to re-explain this again. I had assumed that the authors wanted to highlight that the ASP is a quality initiative with good outcomes in our patients and PAF process is a recommendation In the first paragraph on page, it may be appropriate to re-structure the paragraph to describe PAF better. The same theme occurs throughout the article - the language could be improved upon to get your message across.
3. Page 3 Lines 77-80 - the intent of this paragraph is not clear. The author's message in this paragraph is not apparent.
4. Page 4 Section 2.3 (intervention data)
5. Page 6 (table: the reason for inappropriate meropenem use and subsequent recommendations are linked), to present them on separate lines is repetition of the same information
6. Page 6 figure 2: I think we have to distinguish between 2 points. Unnecessary antibiotics (regardless of whether it is meropenem or not) vs Excessively broad spectrum antibiotics and the labels can be revised
7. Page 6 lines 113 - the authors described a stewardship intervention to switch from meropenem to ertapenem which was well accepted. I was wondering if the ASP team thought about reducing carbapenem exposure, and this theme could be developed further in the discussion.
8. Page 9 - lines 152 - what cultures are the authors referring to? Screening cultures or clinical cultures (blood cultures) to work up a fever
9. Lines 162 - 167 are also difficult to grasp for the reader due to the terminology used or the phrasing. The methods are placed at the bottom of the article but a lot of the structure in the paragraphs on page 9 require information from the methods. please revise.
10. The materials and methods could be presented more succintly
11. The language throughout the article could be improved to help the reader understand the article better.
- Examples.
- Page 2 Figure 1 - Eliminated could have been replaced by excluded
- Carbapenemic - Perhaps we can just use carbapenem
- Page 4 lines 99- "over spectrum" - to replace
- Page 9 - "downgrade meropenem pressure" --> better phrased as "reduce carbapenem pressure"
- "unicentre" --> better described as single center
12. While the review of meropenem audits by the ASP teams is not new, and it is known to be safe and reduce carbapenem consumptions, the prior studies were more heterogenous and this article is relatively "clean in this ASP audits for meropenem". It is also interesting that the authors had results from screening cultures to guide carbapenem recommendations. The data in Table 2 - microbiological data is a bit confusing. What is screening culture, what is clinical culture? I was also wondering if we could correlate the inappropriate antibiotic use to whether is was unncessary or broad and what were the acceptance of these recommendations in light of culture data. The information is hard to glean from what is currently written.
Round 2
Reviewer 1 Report
I am pleased to see that the author addressed all the comments and the manuscript was sufficiently improved.Reviewer 2 Report
Thank you for your revisions.
The main revisions which are still required are linguistical. Please review the paper again.
I will list examples here:
- In the revision in Lines 79-81
"All audited patients with inappropriate meropenem use had at least one risk factor to develop a multiresistant bacterial infection, being then empiric therapy with meropenem correct. The most frequent related factors were the use of antibiotics in the last 30 days and inpatient treatment in the last 60 days (59.4% and 56.5%, respectively); We found no statistical differences between groups."
It could be said that the patients who received meropenem in the study had risk factors for drug resistant bacteria infections, then list the risk factors as you have highlighted in the paragraph above.
2. In lines 85-86, site infection cultures is better described as diagnostic clinical cultures, and this term could be used throughout the manuscript.
3. In the table 2 on Page 5, what does isolation mean?
"Isolation, —n°/total cultures- (%)"
"Screening cultures' isolation, —n°/total"
4. In lines 103-104, it reads better
"The most frequent reason to considerer a prescription inappropriate was the excessively broad use spectrum of meropenem in 66 audits"
5. In line 141, replace the word low with reduce.
6. In lines 224-228, are the authors trying to say that meropenem audits form the bulk of the ASP work?
Just modify the language to improve the clarify of the manuscript.
Thanks.